# Reduction in Robotic Arm Energy Consumption by Particle Swarm Optimization

**Aleš Vysocký** [1,*] , **Richard Papřok** [1] , **Jakub Šafařík** [2] , **Tomáš Kot** [1] , **Zdenko Bobovský** [1] , **Petr Novák** [1] , **and Václav Snášel** [3]

1 Department of Robotics, Faculty of Mechanical Engineering, VSB-Technical University of Ostrava, 70800 Ostrava, Czech Republic; richard.paprok@vsb.cz (R.P.); tomas.kot@vsb.cz (T.K.); zdenko.bobovsky@vsb.cz (Z.B.); petr.novak@vsb.cz (P.N.)

2 Laboratory of Big Data Analysis, IT4Innovations, VSB-Technical University of Ostrava, 70800 Ostrava, Czech Republic; jakub.safarik@vsb.cz

3 Department of Computer Science, Faculty of Electrical Engineering and Computer Science, VSB-Technical University of Ostrava, 70800 Ostrava, Czech Republic; vaclav.snasel@vsb.cz

* Correspondence: ales.vysocky@vsb.cz

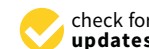

**Featured Application: Proposed method can improve efficiency of non-technological movements of robotic manipulator. This has a wide range of use in the industry, where robot provides traverse between different technological operations.**

**Abstract:** Improved energy usage efficiency is a common goal for economic and environmental reasons. In this manuscript, we present a new approach for the execution of a point-to-point robot motion. The energy efficiency of an industrial or collaborative robot is increased by the reduction of the energy consumption during nontechnological, path-independent movements. The novel trajectory generation method relies on particle swarm optimization with a Bezier curve interpolator. We present the effectiveness of the algorithm on several chosen trajectories, where the best result yields up to 40 % energy saving, while the worst is still at least 10 %. We verified the results of our method by real-world tests on a UR3 robotic arm.

**Keywords:** energy efficiency; manipulators; particle swarm optimization; robot motion; trajectory optimization

## 1. Introduction

We are facing a rapid and constant expansion of the utilization of robots in the industrial world. Together with economic pressure for cost reduction, a growing influence on a sustainable economy (including an effort to control climate change), we can expect emphasis on the decrease of robot operation costs, of which the electricity expenses are a substantial part. Therefore, it is of great importance to search for various ways to improve the energy efficiency of robot operations [1].

A lot of industrial companies can increase their competitiveness and reduce their environmental impacts by improving the energy efficiency of the manufacturing process. Energy costs are a non-negligible part of the total costs for many automated production lines. For example, in the automotive industry, the energy consumption of industrial robots represents approximately 8% of the total energy usage during the production phase [2]. In an automobile production plant, around two-thirds of the power demand is accounted for by electric motors that drive conveyor belts, machinery, and pumps, or that operate robotic joints. The energy consumption of large industrial robots can be enormous, even 100 kWh per day. For a large automatized manufacturing factory, the total energy used is comparable to that of a mid-sized town.

The reduction of robot energy consumption is possible by the development of new robots by hardware optimization, e.g., with lower weight part, higher stiffness, higher efficiency of their drives, and advanced energy recuperation systems, etc. [3]

The total number of existing industrial robots is approximately 2.7 million worldwide [4]. It is possible to optimize their operation, e.g., by a decrease of machine idle time, by a reduction of power consumption during this idle time, or by trajectory optimization [3].

Nowadays, almost no industrial robots have optimized movements and little attention has been devoted to improving efficiency by optimizing robotic movement. Most of the optimizations only aim at the production rate. There are examples of energy consumption optimization for several cooperating robots of a robotic cell given in [5,6].

This paper presents a method for energy usage reduction by optimization of motion trajectory, which enables optimized energy movements and saves approximately 10 % of robot energy consumption. The method that is not excessively complicated for implementation, and it can be done, in principle, by an averagely skilled technician is a good option for energy consumption reduction.

The manuscript is organized, as follows: in Section 2, we give a short overview of the methods that are used in point-to-point (PTP) optimization and extract some theory. In Section 3, we describe our approach and then, in Section 4, we present the achieved results on the robotic arm accompanied by discussion. In Section 5, we summarize our work and outline the direction of future work.

## 2. Point-to-Point Energy Optimization

The proposed method demonstrates the use of particle swarm optimization (PSO) in the robotic field for PTP trajectory energy optimization.

There are several papers that are devoted to PTP optimization, each targeting different goals—many of them aiming at minimal task operation time. However, time is not always an available objective to optimize, especially in specific cases where the arm waits for the completion of a task performed by another agent, gluing, or welding. We focus this manuscript on the energy optimization of possible fixed time trajectories having specific start and end positions. This kind of task represents a typical arm movement during a pick and place job. We only limit the scope of the following research to energy optimization for a robotic arm with six degrees of freedom (DOF) in PTP tasks.

Various authors search for energy-efficient path planning and energy-saving methods for robotic arms. There are detailed and depth reviews [3,7] on various optimization aspects in the field of robotic manipulators.

Using the calculus of variations, one can solve the PTP energy optimization as shown in [8,9]. A simple example of such a problem is to find the curve giving the shortest length for the PTP connection. If there are no constraints, the solution is a straight line between the points. However, if the curve is constrained to lie on a surface in space, then the solution is less obvious, and possibly many solutions may exist. Such solutions are known as geodesics. The topology of PTP connection by arm is more complex, and direct computation of geodesics is very difficult [10,11].

In [12], the authors solve the energy optimization of a manipulator with the use of a genetic algorithm (GA), where the chromosome represents the arm trajectory, as $n$ points in the joint space. Research by Sengupta et al. [13] solves the energy optimization for straight, square, and combined motion using robot simulation software, while taking into consideration velocity, acceleration, and trajectory. The optimization module then iteratively evaluates all of the solutions and chooses the one with the lowest energy demand. Paryanto [2] measures the total robot energy consumption of the whole manipulator, the control unit, and robotic arm, whlie using an external measurement unit. The simulations and optimization rely on dynamic behavior modeling in Catia software. The optimization key elements are robot accuracy, productivity (execution time), and power consumption reduction.

He et al. [14] compare the energy optimization results from GA and Constraint Immune multiobjective optimization algorithm. They present the trajectories as eight-point cubic B-Spline curves.

Paes and others [15] optimize the energy and time of the trajectories of robotic arms from ABB Corporation, but the producer's policy restricts the use of technical data. The authors of [15] optimize a path with sequential quadratic programming while using a Matlab model that they directly obtain from proprietary software from ABB.

Further information on the topic of PTP trajectory energy optimization can be found in review papers [16] in Chapter 4, [3] in Chapter 3, or [17].

All of the papers dealing with PTP energy optimization share similar features. First, there are inverse and direct approaches to optimization. For the inverse approach previously discussed in [18–20], the researchers develop a mathematical model for the theoretical prediction of power consumption during the execution of particular trajectories. Subsequently, they integrate the data to obtain the theoretical energy consumption estimation over time.

Even for a simple movement, there are numerous possible trajectories for an exact specification of the movement. The analytical models vary in complexity and need to handle various aspects of robotic arm movement for all six DOF. The basic are the kinematic and dynamic factors of the manipulator: length and mass of the links, inertia, gear ratio, friction, motor torque, winding resistance, robotic parts, temperature, etc. Subsequently, there are applied forces: gravitational, centrifugal, Coriolis, etc. The analysis of the preciseness of these models can be found in [2].

Another way to express energy efficiency is by simplification of the problem to a single optimization criterion, such as minimum effort, the time integral of squared or absolute values of torque, squared joint accelerations, see [21,22]. However, for the price of seamless simplification and decreased precision.

Using the direct approach, we can measure the energy consumption value right from the real robotic arm. Unlike for the inverse approach, this method depends on a specific robotic arm and its hardware and software to enable such a measurement. There is also the possibility to measure the overall power consumption from a tool that was connected to the manipulator power cables [3].

Another challenge lies in the representation of the movement trajectory with given constraints. Because of the difficulties in Cartesian space representations (singularities, rapid joint movements, etc.), most of the authors prefer the joint space representation. We can define the trajectory itself by a set of points in the joint space, uniform cubic B-splines [18,19], Bezier curves [23], etc.

The commonly used optimization techniques in the field of robotic are GA, exhaustive search, Differential evolution (DE), Firefly algorithm (FA), Artificial Immune system (AIS), PSO, etc. Complete review of robotic arm optimization algorithms is beyond the scope of this paper. Figure 1 shows the average performance of several algorithms in the task of robotic arm torque optimization.

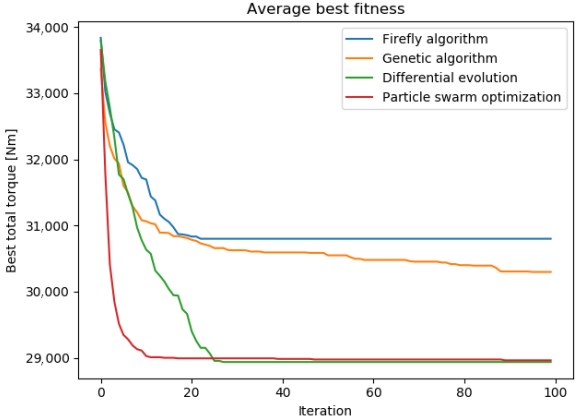

**Figure 1.** Average best fitness for torque optimization task

The algorithms were tuned for the task and their performance was evaluated in 20 runs. The best possible value of torque (overal sum of torques of all joints during the whole movement) was found

by DE and PSO. Moreover, the PSO algorithm found the best value in the least number of iterations. Therefore, we will use PSO for the following energy optimization task because of the performance in the torque optimization task.

## 3. Description of the Method

### 3.1. Particle Swarm Optimization

The PSO is a derivative-free iterative stochastic optimization algorithm based on swarm theory and observation of social behavior of animals, e.g., bird flocks or fish school [24]. This method suits our purpose, because we will not know the energy consumption function and we do not try to construct it for our particular case of the Universal Robot robotic arm (model UR3)–Figure 2.

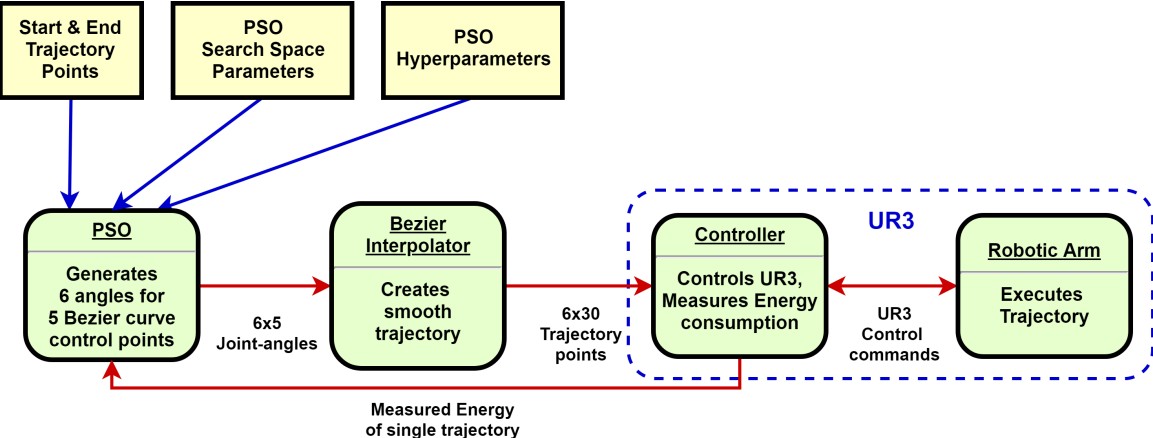

**Figure 2.** Block diagram of the proposed algorithm

The population contains candidate solutions, called particles. Particles are initialized with random positions that correlate with a solution to the problem. Each particle has a random velocity vector defining its movement through a solution space. The individual particle then dynamically adjusts its velocity and searches the solution space according to its own movement experience, the same as the best solution from the rest of the population. All of the particles are grouped into a swarm and cooperate on finding the best solution within the search space.

In our work, we use the PSO algorithm with cognitive and social components. The particle positions in our method serve as control points. The solution space is defined as a $J$-dimensional area containing a population of $I$ particles. The particle $i$ is then located at the position $X_i = x_{i1}, x_{i2}, \ldots, x_{ij}$ with a velocity $V_i = v_{i1}, v_{i2}, \ldots, v_{ij}$. In each iteration of the PSO algorithm, the particle updates its position and then moves randomly towards its best-known position ($L_i$), denoted as $L_i = l_{i1}, l_{i2}, \ldots, l_{ij}$ and to the best position known in the swarm $G_{best} = g_1, g_2, \ldots, g_j$. Particle positions are calculated as the addition of the actual velocity to the previous position:

$$X_i^{t+1} = X_i^t + V_i^{t+1} \tag{1}$$

Equation (2) shows the computation of the particle velocity:

$$v_{ij}^{t+1} = w * v_{ij}^t + c_1 r_1 [l_{ij}^t - x_{ij}^t] + c_2 r_2 [g_j^t - x_{ij}^t] \tag{2}$$

where $v_{ij}^t$ is the velocity of particle $i$ in dimension $j$ at time $t$. The inertial weight $w$ influences the velocity from the previous iteration to control the exploration and exploitation of the search space. A small $w$ urges the particle to exploit the area in its near position, while a high value prevents the particle from being stuck in a local minimum [25].

Values $r_1$ and $r_2$ are independent random numbers from a uniform distribution over a range from 0 to 1. The constants $c_1$ and $c_2$ affect the acceleration that pulls the particle to the best position from the particle memory and the best-known particle position from the swarm. In other words, $c_1$ influences the cognitive part of the particle, representing primitive thinking. The constant $c_2$ takes part in social behavior and it represents the cooperation among particles.

Various recommendations about the values of $c_1$ and $c_2$ exist. Some authors recommend values over the range from 0 to 1, and the sum of $c_1$ and $c_2$ is less than or equal to 1 [26,27]. Engelbrecht and others [25,28] use inertial weight in Equation (3) to show that particles have convergent trajectories, even for values of $c_1$ and $c_2$ higher than 1.

Liang et al. [29] proposed PSOs with a new comprehensive learning strategy (CLPSO), where each dimension of a particle learned from just one particle best historical information, while each particle learned from a different particles best historical information for different dimensions.

$$w > \frac{1}{2}(c_1 + c_2) - 1 \qquad (3)$$

By Engelbrecht, cyclic or divergent particle trajectories may occur when Equation (3) is not satisfied [25]. Other researchers [30] drew similar conclusions.

The termination criterion for basic PSO is reaching a specific generation, the designated value of $G_{best}$, or its saturation. One generation means updating all positions of particles, computation of the fitness function, final evaluation, and update of their local $l_{ij}$ and global $G_{best}$ best. The setting of PSO control parameters was found while using empirical testing and are summarized in Table 1. The algorithm will keep good performance and enough particle divergence, preventing from getting stuck in local optima. The choice of parameter values is similar to research from Engelbrecht [25,31,32].

**Table 1.** PSO parameters involved in all runs.

| Parameter(s) | Value |
| --- | --- |
| Max. number of PSO iterations | 250 |
| Particles in swarm | 30 |
| Number of dimensions | 30 |
| Inertia weight ($w$) | 0.810 |
| Search space boundary | $[-6.28, 6.28]$ |
| Velocity boundary ($\Delta_1, \Delta_5$) | $[-0.1, 0.1]$ |
| Velocity boundary ($\Delta_2, \Delta_3, \Delta_4$) | $[-0.2, 0.2]$ |
| Local weight ($c_1$) | 0.495 |
| Global weight ($c_2$) | 0.724 |

There are a lot of proposed techniques for the improvement of PSO performance. They target different aspects of the algorithm: configuration parameters, learning strategies (for updating of positions), swarm diversity, etc. In [33], the authors propose a new method for interactive learning. Their approach is effective against getting stuck in a local minimum, especially for multimodal problems.

Some papers combine PSO with other evolutionary algorithms and create hybrid PSO. The authors in [34] combine conventional PSO with an estimation of distribution (EDA) algorithm. EDA enhanced the performance of the hybrid PSO algorithm when compared with other PSO variants on the IEEE Congress on Evolutionary Computation 2003 (CEC2003) testing framework. However, their solutions bring higher computational requirements, as it adds more complex PSO behaviour and generates new particles during computation.

Xu et al. [35] proposed two-swarm learning PSO (TSLPSO), which is a PSO variant with subpopulations. One uses the dimensional learning strategy for fast convergence to the global best

position, while the second subpopulation relies on a comprehensive learning strategy and maintains high particle diversity in the search space. Although this approach gives significant improvements in performance on various test functions, our real problem brings TSLPSO high overhead from an increased number of particle fitness evaluations.

We were strictly aiming for a fast convergence to the $G_{best}$ value, while keeping a low number of fitness function evaluations. In our preliminary research, there was not a signification boost in performance while adopting any of the aforementioned PSO techniques.

### 3.2. Trajectory Generation

We parametrize a set of admissible trajectories by Bezier curves in joint angle space, which is free of singularities from arbitrary movements. The reason for the choice of Bezier curves was motivated by the fact that the direct use of PSO points as node points of a trajectory generate very curved trajectories that frequently led to abrupt motions and protective stops of the robotic arm. Bezier curves serve as a smoothing mechanism. These curves have low curvature and smooth trajectories with continuous derivatives of all orders and are, therefore, compatible with our goal to search for trajectories with minimal energy consumption.

Bezier curves are defined by $n + 1$ control points: $P_0, P_1, \ldots, P_n$, with the following equation:

$$C(u) = \sum_{i=0}^{n} B_{n,i}(u) P_i \tag{4}$$

where $B_{n,i}$ are Bernstein basis polynomials of degree $n$, defined as:

$$B_{n,i}(u) = \frac{n!}{i!(n-i)!} u^i (1-u)^{n-i} \tag{5}$$

At the beginning of our experiments, we tried optimization with only one waypoint in the middle of the trajectory. This simple approach led to an energy consumption reduction. However, the extension of the searchable trajectory space that was controlled by seven control points in the robotic arm six-dimensional joint space led to a further improvement of the energy consumption.

Two of the seven Bezier curve control points are the start and end points of the trajectory and they are not subject to optimization. The next two control points in the immediate vicinity of the start and end points are optimized by the PSO algorithm over a smaller range in order to avoid abrupt starts and stops of movement, which could cause a protective stop of the robotic arm. In other words, we demand that the proposed trajectories have smaller time derivatives in the endpoints, which physically means that they have a lower velocity close to the endpoints.

The last three control points are used for proper energy optimization of the trajectory and they are responsible for the variation of the shapes of PSO proposed trajectories. See Figures 3 and 4 for a graphical explanation of this point.

Control points of Bezier curves are generated in specific ranges $\Delta_1 - \Delta_5$, which serve as PSO search space limits (see Figure 3 and Table 1). The control points regulate the size and shape of the joint angle trajectory space $\theta(t)_{0-5}$ of robotic arm movements. In our experiments, we chose such a range that the arm could not hit any obstacle—we ensured enough free space around the manipulator to avoid any collision. In real applications, a proper collision detection would be necessary during the PSO algorithm (penalizing particles that represent a trajectory with some collisions).

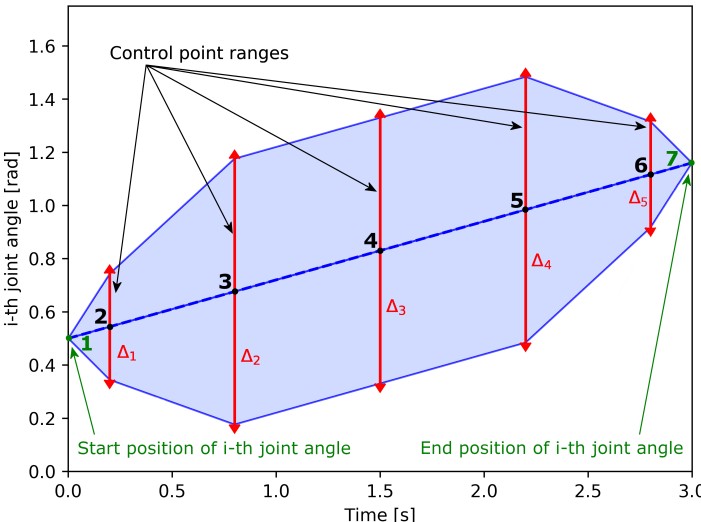

**Figure 3.** PSO search space for the i-th joint angle. The space is given by Bezier control points in the blue area. The blue area is a convex hull of all admissible PSO trajectories.

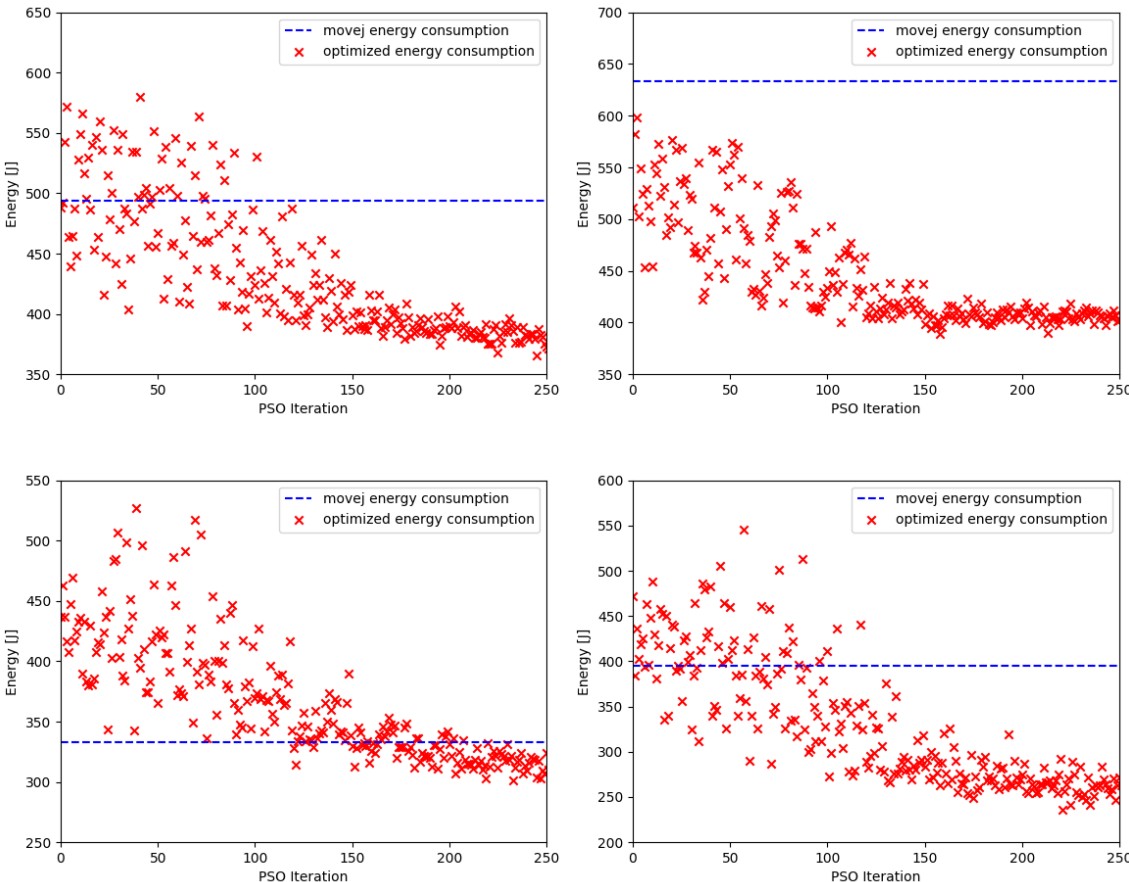

**Figure 4.** Convergence of particle swarm optimization (PSO) optimizations. The blue line represent default energy consumption by individual PTP movement, red crosses are energies consumed by execution of optimized trajectories.

The maximum velocity of individual joints is set by the steepest line segment of the blue area in Figure 3, i.e., the line with the highest absolute value of the time derivative. Robotic arm

manufacturers usually provide information of the maximal achievable velocity of individual joints. Another mechanism is to set this by trial and error.

Maximum acceleration control was performed by manual adjustment of the control points 2 and 6, see Figure 3. In the uncontrolled case, reaching the end position sometimes led to a too fast deceleration, which invoked the UR3 protective stop. Another mechanism for acceleration control would be the computation of second derivatives from known formulas of the Bezier curve second derivatives. This would involve more manual work and lower practicability, especially in the case when one is not aware of the safe values of maximum acceleration.

## 4. Experiment Setup And Results

### 4.1. Ur3 Robotic Arm

UR3 (Figure 5b) is a six-degrees-of-freedom collaborative robot with the maximum payload of 3 kg and 500 mm reach. The motion controller of the robot is easily accessible with an external computer and it can also be replaced with a custom one. In this work, the standard motion controller with predefined commands was used. The controller supports commands for interpolation in Cartesian space (movec–arc segment, movel–line segment), and movej for interpolation in joint space.

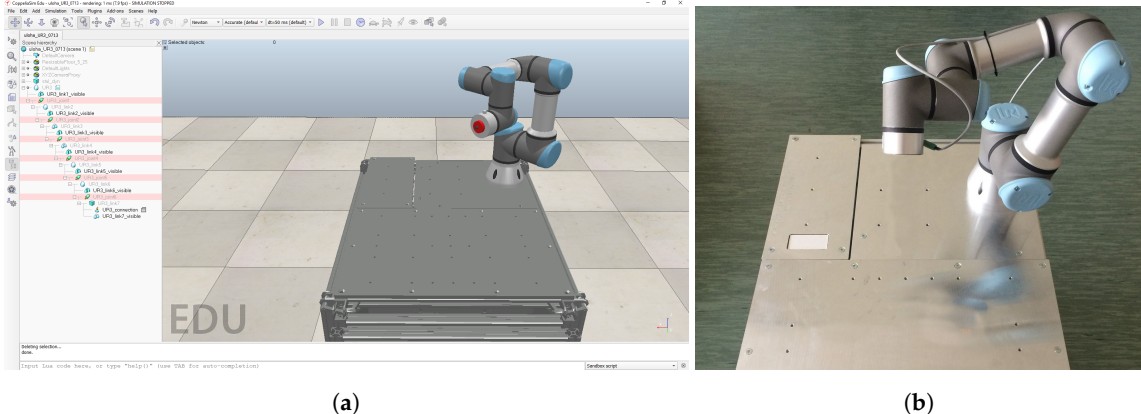

|  |  |
|:---:|:---:|
| (**a**) | (**b**) |

**Figure 5.** Experimental measurement setup: (**a**) Simulation environment in CoppeliaSim, (**b**) UR3 robot mounted to the table.

For the experiment, we chose arbitrary collision-free point-to-point movements. Using the movej command, the trajectory generation is based on path planning with trapezoidal timing. This provides a smoothly interpolated movement of all joints with the same duration. Trajectory that was generated with this command was used as a benchmark to compare the efficiency with the suggested optimization method.

In order convert the optimized trajectory generated as a set of position data into a robot movement, the movej command is not appropriate, because it is a command for rest-to-rest or blend-connected movements. For real-time control, the servoj and speedj commands are needed. The speedj command offers velocity control and provides a trajectory without exceeding the maximum limits for acceleration and jerk during movement. Based on the input data, we used the servoj command, which can move the robot to the desired joint position configuration with an input update rate of 8 ms.

Figure 6 shows a comparison of the measured angle of the 2nd joint (shoulder)—this joint was chosen because it lifts the weight of the whole arm. The movej command shows a smooth change of the angle, while a set of servoj commands with different timing frequencies report a deviation with an amplitude of approximately one degree (0.02 rad). By comparing the frequencies, we can see that there is not a significant difference, except for the measurement with 8 ms timing, which is closer to the movej data. We applied a floating average in the graph to filter the noise from the measurements. Figure 7 shows a measurement of the electric current, which is the base for computing the energy

consumption. The measured currents are very similar during the whole movement, even while using lower frequencies, which assures results similar to the movej command. The shortest timing (8 ms) is on the edge of the communication capabilities of the robot and it causes some issues with the stability of communication, especially during longer movements. Thus, we decided to avoid this timing.

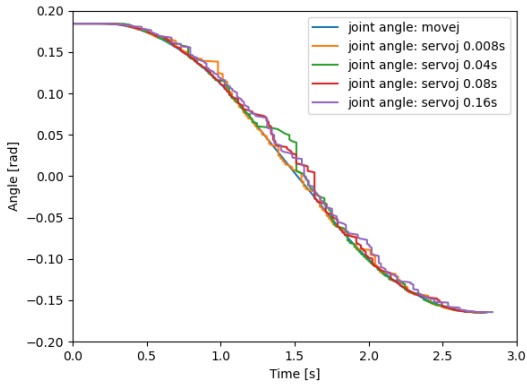

(**a**) Measured position during the movement

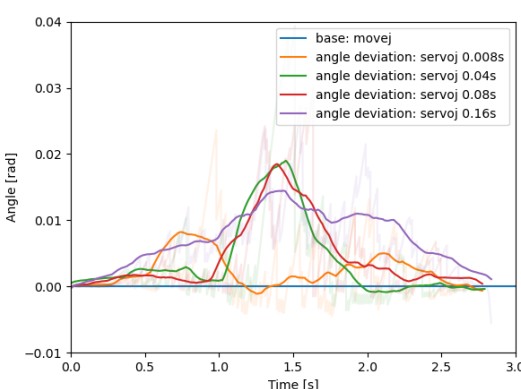

(**b**) Position difference of movej and a set of servoj

**Figure 6.** Position deviation of the 2nd joint (shoulder) between measurements with different movement prescription during the movement A (Figure 8a).

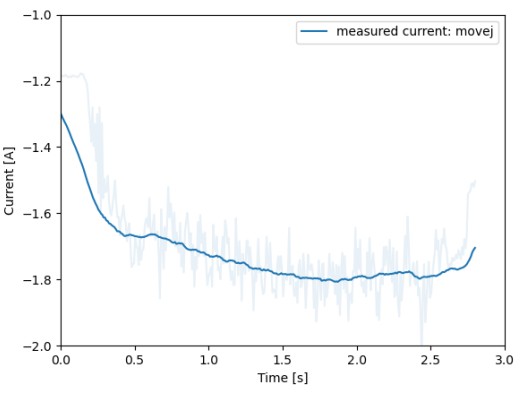

(**a**) Measured electric current during movement

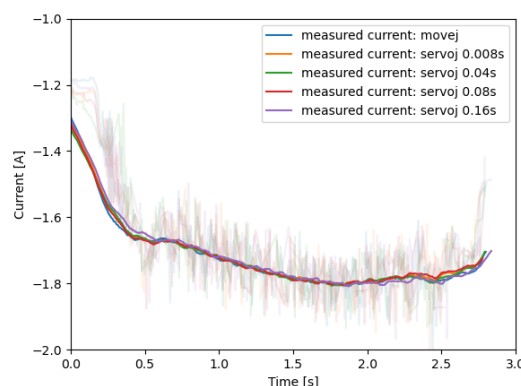

(**b**) Current difference of movej and a set of servoj

**Figure 7.** Electric current deviation of the 2nd joint (shoulder) between measurements with different movement prescription during the movement A (Figure 8a).

The desired trajectory generated with the Bezier curve based optimization provides position data for the servoj command. The control program consists of a set of servoj commands with a time step of 80 ms. There is a time delay before the robot starts a movement, which is caused by a shift between the measured data and movej benchmark data. The recording is postponed until a movement is detected. To reach the full dynamic performance of the robot, it is necessary to disable the collaborative restrictions and set the maximum operation parameters; otherwise, the measured data might be distorted.

For communication with the internal motion controller, we used a TCP/IP real-time client that supports a 125 Hz update rate. The client communicates with the manipulator via the URX library. We made slight modifications to this library for the instant reading of joint currents. The real-time data exchange module is used in order to collect the actual state of the manipulator. The URX library provides a preset connection setup and the most important commands to control the robot and obtain

the actual state. To measure the energy, it is necessary to decode and evaluate the joint voltage and actual currents from the robot state messages. Energy is then computed, as follows:

$$E = \sum_{joints} \int I(t)V(t)dt \qquad (6)$$

The energy measurement gives a noisy signal, which has at least five causes: network issues (packet loss and network jitter) in a real-time communication, random vibration of the robot base and table due to a high acceleration, temperature of robot parts, gearbox lubricant conditions, and joint friction.

### 4.2. Tested Movements

We have chosen four representative arm movements and investigated how much energy can be saved compared to baseline movements made by the movej commands, which produce linear curves in the joint space blended by a set acceleration of nearby endpoints. The chosen movements represent four categories of typical motions:

- A—manipulation from side to side (Figure 8a) represented by a left-to-right motion with the rotation angle of 90° and the tool centre point (TCP) staying at approximately the same height,
- B—manipulation from front to back (Figure 8b) represented by a left-to-right motion with the rotation angle of 180° close to the robot base,
- C—manipulation from bottom to top (Figure 8c) represented by a movement between the horizontal position and the upright position, and
- D—manipulation from top to bottom (Figure 8c) represented by a movement between the upright position and the horizontal position.

The duration of all the movements is approximately 3 s, so that the results can be reasonably compared by the energy consumption.

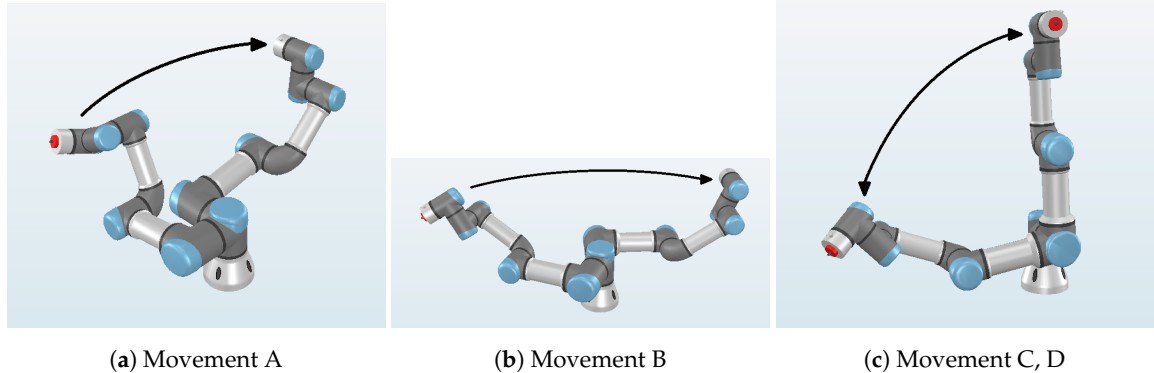

| (**a**) Movement A | (**b**) Movement B | (**c**) Movement C, D |

**Figure 8.** End positions of the optimized trajectories.

As already mentioned above, we have not implemented a collision avoidance mechanism in our setup—our testing conditions provide enough free space around the robotic arm and the purpose of the work is just the verification of the proposed general principle. In cases when one would have to deal with a limited working area with a danger of collisions, we recommend setting the objective function (energy used for the PSO proposed movement) to a high value of energy for the case of a self-collision or a collision with the environment. This approach would also give a successful optimization.

### 4.3. Results

In our experiments, we observed a decrease in energy consumption of the chosen movements in the interval from 10 % to almost 40 %, as can be seen in Table 2. The optimization used a PSO

swarm size of 30 particles and required up to two hundred iterations (eras) for this result. We compare measured electric current in individual joints during movej movement and optimized movement in the Figure 9. The physical reason for the energy reduction lies, at least partly, in finding a path during which the center of mass of the robotic arm is on average closer to the vertical axis passing through the robot base.

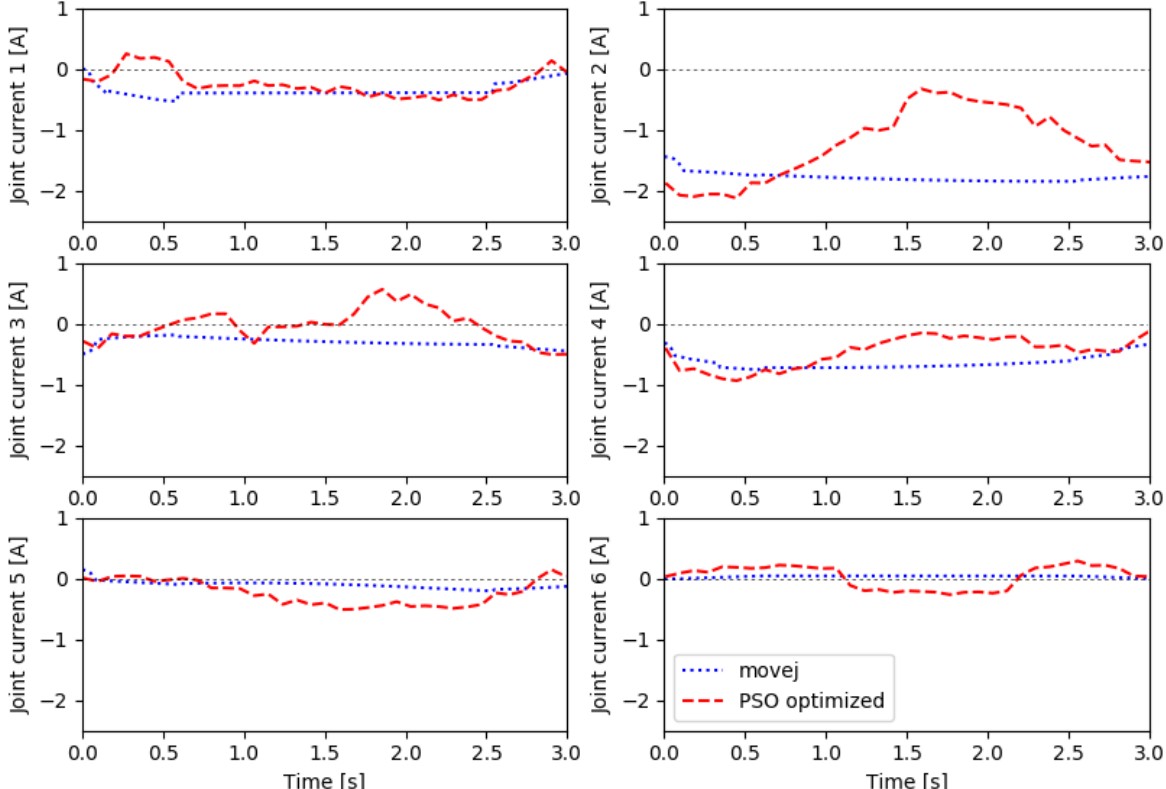

**Figure 9.** Joint currents for movement A : the optimized trajectory (including control points of Bezier curves) and trajectory obtained by execution of default movej command on UR3.

**Table 2.** Table of energy consumption for given movements.

| Movement Type | Energy Consumed (Mean) | | Savings |
|---|---|---|---|
| | Movej [J] | optimized [J] | |
| Movement A | 495 | 377 | 23% |
| Movement B | 634 | 400 | 37% |
| Movement C | 340 | 305 | 10% |
| Movement D | 400 | 335 | 16% |

In Figure 10, we can see that it would be difficult to program such a movement by hand. Joint 1 remains at the beginning of the movement in the initial position, while joint 2, which is mainly responsible for the reduction of torque due to gravity, goes closer to the vertical axis that intersects the robot base. After this decrease of momentum in inertia, lower energy consuming rotational acceleration in joint 1 is performed. Similarly, joint 4 in the optimized case goes to the final position faster than in the reference case, because it leads to a faster route for more momentum and energy-efficient position.

Interestingly, the energy optimization with proper total energy consumption computation, i.e., either taken from a reasonable theoretical model or real measurements (our case), leads to a $\dot{\theta}_i(t) \sim 0$ on the trajectory end. Accordingly, this is a natural result of energy optimization, unlike many

cases of energy optimization methods, where we first mathematically enforce zero joint velocities at the end of the trajectory by hand, and we then optimize the trajectory. It is a natural result of the energy optimization process that there is a soft start and a slowdown at the end of the trajectory. For other cases, movements B and C, we have observed similar tendencies to undergo such energetically reasonable behavior during the optimization process.

During the optimization process, one can see that the robot arm movements converge to movements that resemble more and more natural human-like motions, which we assume to be energetically optimized.

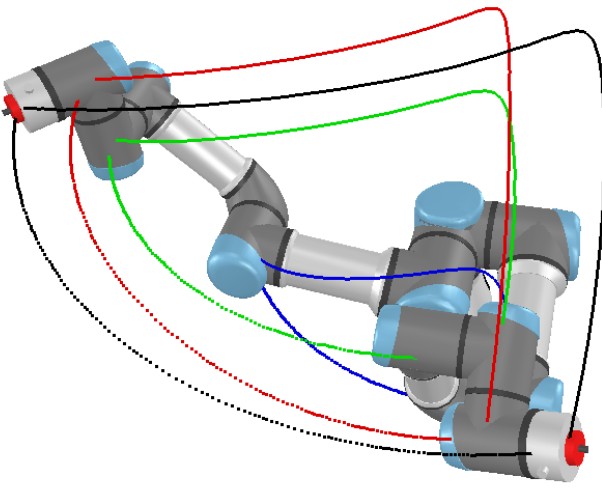

**Figure 10.** Comparision of movej (dotted line) and PSO optimized (solid line) joints and tool centre point (TCP) trajectories; elbow is represented by blue color, wrist joints with red and green and TCP trajectory is depicted with black line.

## 5. Conclusions, Discussion, and Future Work

We have proposed a novel method for reducing energy consumption in repeated robotic arm tasks. The energy saving in comparison to a default trajectory generated by a controller (standard trapezoid movement) in our test cases ranges from 10% to almost 40%. The necessary values were directly acquired from the robotic arm, without the need of a complex power consumption model. The method relies on the PSO algorithm. In combination with a Bezier curve generator, the proposed method generates smooth and energy efficient trajectories for point-to-point movements. The proposed method is applicable to robotic arms with the same kinematic structure (6 DOF). It is possible to generalize this method to a manipulator with arbitrary kinematic structure.

Another benefit of this method is the possibility to quickly find the optimal trajectory under a different objective metric by using a proper fitness function, e.g., minimum acceleration, jerk, torque, etc. Because the method does not use higher polynomial trajectory generation, the computation has fewer limitations (like zero derivations, restrictions for start or end points) and it may be able to search a wider solution space.

The potential weakness of this approach is in the necessity of performing several hundreds of real-world trials to find the optimal trajectory. However, there was a relatively fast convergence to the optimal trajectory in most experiments (see Figure 4). With a precise dynamic model, it would be possible to find the optimal trajectory without the need for any real-world movement execution. In order to achieve this, we have also tried simulations of movements in the simulation system CoppeliaSim (Figure 5a). However, the energy prediction from the CoppeliaSim tool does not perfectly correspond to real-world measurements. In our future work, we will try to find a way to lower the error of predicted energy consumption and remove the dependency on real robotic arm trials.

The proposed method does not implement any collision detection or avoidance mechanism and it requires the proper setting of velocity boundaries on the robotic arm. Moreover, we did not use any payload or end-effector. The size of the payload would change the energy consumption. We intend to investigate the influence of payload (or the ratio of payload) concerning the robotic arm mass distribution in the future.

Experience that was gathered during this work suggested some interesting topics for future work. The most challenging is the possibility to train a model for power consumption with a feed-forward or recurrent neural network. If successful, this would remove the need for time-consuming real-world trials while still enabling an accurate solution and the possibility to quickly find optimized trajectories with different endpoints. The robotic arm manufacturer could deliver such a neural network or one could be produced by the robotic community to further ease the energy usage and make more efficient robot motion. Another topic would be the use of reinforcement learning. The shaped reward function with energy consumption term could be led to a practical adoption of energy saving methods for robotic movements.

**Author Contributions:** Conceptualization, R.P. and Z.B.; methodology, R.P.; software, R.P.; validation, A.V., J.Š. and T.K.; formal analysis, A.V.; resources, A.V.; data curation, J.S.; writing—original draft preparation, R.P. and A.V.; writing–review and editing, T.K. and P.N.; visualization, A.V.; supervision, Z.B.; project administration, V.S.; funding acquisition, P.N. All authors have read and agreed to the published version of the manuscript.

**Funding:** This work was supported by the European Regional Development Fund in the Research Centre of Advanced Mechatronic Systems project, project number CZ.02.1.01/0.0/0.0/16_019/0000867 within the Operational Programme Research, Development and Education and by the European Regional Development Fund under the project AI&Reasoning (reg. no. CZ.02.1.01/0.0/0.0/15 003/0000466).

**Conflicts of Interest:** The authors declare no conflict of interest.

## Abbreviations

The following abbreviations are used in this manuscript:

| | |
|---|---|
| ABB | Asea Brown Boveri corporation |
| CEC2003 | IEEE Congress on Evolutionary Computation 2003 |
| CLPSO | Comprehensive learning particle swarm optimization |
| DOF | degrees of freedom |
| EDA | Estimation of distribution algorithm |
| GA | Genetic algorithm |
| PSO | Particle swarm optimization |
| PTP | Point-to-point trajectory |
| TSLPSO | Two swarm learning particle swarm optimization |

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
