# Peer review of "Reduction in Robotic Arm Energy Consumption by Particle Swarm Optimization"

_applsci, doi:10.3390/app10228241_

Round 1
Reviewer 1 Report
The paper is written well and can be accepted to publish in the Journal.
The authors could add some fundamental theories to the paper about the PSO and bezier curve.
Author Response
The authors appreciate all the reviewers’ comments. Thank you for the careful reading of our manuscript and your valuable suggestions. After considering the comments carefully, we have revised the manuscript accordingly. All changes (including the changes added in reaction to the other reviewers) are highlighted in yellow in the article.
On the line 88 and 106 we have added justification for using the PSO algorithm and a comparison of PSO with other optimization techniques.
Reviewer 2 Report
The manuscript deploys a Particle Swarm Optimization based technique to minimize energy consumption of a robotic arm and demonstrate their method on a UR3 robotic arm.
The paper would benefit from further motivation on why PSO is a suitable method for energy optimization for complex robotic systems. The UR3 robotic arm and similar industrial robots are extremely well-studied and their inertial properties can be specified to a high degree of certainty that suggests analytical solutions would be promising, especially considering the small set of target motions described in Section 4.2.
Section 4.3: Have you studied whether there's any inherent difference in energy consumption between movej and servoj based motion? How do the experimental results compare when different command frequencies are used with servoj?
Overall, I would recommend extending the experiments section with significantly more data on diverse motions, and potentially demonstrate that the algorithm yields similar benefits over different robots as well.
Author Response
The authors appreciate all the reviewers’ comments. Thank you for the careful reading of our manuscript and your valuable suggestions. After considering the comments carefully, we have revised the manuscript accordingly. All changes (including the changes added in reaction to the other reviewers) are highlighted in yellow in the article.
Reviewer:
The paper would benefit from further motivation on why PSO is a suitable method for energy optimization for complex robotic systems. The UR3 robotic arm and similar industrial robots are extremely well-studied and their inertial properties can be specified to a high degree of certainty that suggests analytical solutions would be promising, especially considering the small set of target motions described in Section 4.2.
Answer:
Analytical solutions have significant benefits when one has a precise information about inertial properties. But this is the weak point of the measurements with robots. Either there is no perfect information from the manufacturer or every manipulator is slightly different and must be calibrated. Custom changes such as additional periphery, cables, and hoses mounted to the robot disrupt the analytical solution. We added a comment on the line 88. Solution with PSO might be adapted to the customized solution.
Reviewer:
Section 4.3: Have you studied whether there's any inherent difference in energy consumption between movej and servoj based motion? How do the experimental results compare when different command frequencies are used with servoj?
Answer:
We have added a paragraph on the line 228 which describes our solution of substitution of standard interpolation in joint space (movej) with a set of servoj commands. The results are shown in Figures 5 and 6. We compared the movej movement with different servoj sets with different timing.
Reviewer:
Overall, I would recommend extending the experiments section with significantly more data on diverse motions, and potentially demonstrate that the algorithm yields similar benefits over different robots as well.
Answer:
Because we used the experimental measurement as a proof of our theory, we implemented the optimization process to the device we have in our laboratory. We performed 3 sample movements which include different position changes, changes of rotation and different joint behavior. Demonstration on different robots would require the access to the robot, creating new interfaces to control and measure the robot and performing a huge amount of measurements, which is not feasible for us now.